# Design, Synthesis, In Silico and In Vitro Studies of New Immunomodulatory Anticancer Nicotinamide Derivatives Targeting VEGFR-2

**DOI:** 10.3390/molecules27134079

**Published:** 2022-06-24

**Authors:** Reda G. Yousef, Wagdy M. Eldehna, Alaa Elwan, Abdelaziz S. Abdelaziz, Ahmed B. M. Mehany, Ibraheem M. M. Gobaara, Bshra A. Alsfouk, Eslam B. Elkaeed, Ahmed M. Metwaly, Ibrahim H. Eissa

**Affiliations:** 1Pharmaceutical Medicinal Chemistry and Drug Design Department, Faculty of Pharmacy (Boys), Al-Azhar University, Cairo 11884, Egypt; redayousof@azhar.edu.eg (R.G.Y.); alaaelwan34@azhar.edu.eg (A.E.); zi.17.sh@gmail.com (A.S.A.); 2Department of Pharmaceutical Chemistry, Faculty of Pharmacy, Kafrelsheikh University, Kafrelsheikh 33516, Egypt; wagdy2000@gmail.com; 3Zoology Department, Faculty of Science (Boys), Al-Azhar University, Cairo 11884, Egypt; abelal_81@azhar.edu.eg (A.B.M.M.); ibraheemgobaara@azhar.edu.eg (I.M.M.G.); 4Department of Pharmaceutical Sciences, College of Pharmacy, Princess Nourah bint Abdulrahman University, P.O. Box 84428, Riyadh 11671, Saudi Arabia; baalsfouk@pnu.edu.sa; 5Department of Pharmaceutical Organic Chemistry, Faculty of Pharmacy (Boys), Al-Azhar University, Cairo 11884, Egypt; eslamkaeed@azhar.edu.eg; 6Pharmacognosy and Medicinal Plants Department, Faculty of Pharmacy (Boys), Al-Azhar University, Cairo 11884, Egypt; 7Biopharmaceutical Products Research Department, Genetic Engineering and Biotechnology Research Institute, City of Scientific Research and Technological Applications (SRTA-City), Alexandria 21934, Egypt

**Keywords:** anticancer, immunomodulatory, apoptosis, in silico studies, nicotinamide, VEGFR-2

## Abstract

VEGFR-2, the subtype receptor tyrosine kinase (RTK) responsible for angiogenesis, is expressed in various cancer cells. Thus, VEGFER-2 inhibition is an efficient approach for the discovery of new anticancer agents. Accordingly, a new set of nicotinamide derivatives were designed and synthesized to be VEGFR-2 inhibitors. The chemical structures were confirmed using IR, ^1^H-NMR, and ^13^C-NMR spectroscopy. The obtained compounds were examined for their anti-proliferative activities against the human cancer cell lines (HCT-116 and HepG2). VEGFR-2 inhibitory activities were determined for the titled compounds. Compound **8** exhibited the strongest anti-proliferative activities with IC_50_ values of 5.4 and 7.1 µM against HCT-116 and HepG2, respectively. Interestingly, compound **8** was the most potent VEGFR-2 inhibitor with an IC_50_ value of 77.02 nM (compare to sorafenib: IC_50_ = 53.65 nM). Treatment of HCT-116 cells with compound **8** produced arrest of the cell cycle at the G0–G1 phase and a total apoptosis increase from 3.05 to 19.82%—6.5-fold in comparison to the negative control. In addition, compound **8** caused significant increases in the expression levels of caspase-8 (9.4-fold) and Bax (9.2-fold), and a significant decrease in the Bcl-2 expression level (3-fold). The effects of compound **8** on the levels of the immunomodulatory proteins (TNF-α and IL-6) were examined. There was a marked decrease in the level of TNF-α (92.37%) compared to the control (82.47%) and a non-significant reduction in the level of IL-6. In silico docking, molecular dynamics simulations, and MM-PBSA studies revealed the high affinity, the correct binding, and the optimum dynamics of compound 8 inside the active site of VEGFR-2. Finally, in silico ADMET and toxicity studies indicated acceptable values of drug-likeness. In conclusion, compound **8** has emerged as a promising anti-proliferative agent targeting VEGFR-2 with significant apoptotic and immunomodulatory effects.

## 1. Introduction

Cancer is a pernicious disease characterized by uncontrolled and overexcited cell differentiation and division, along with the possibility of such cells invading other parts of the body, leading to death [1]. While progress has been made in treatment and prevention, the global burden of cancer mortality is increasing tremendously. Epidemiological studies revealed the responsibility of cancer for one-fifth of all deaths [1]. In 2020, more than 19 million new cancer cases were estimated, and almost 10.0 million deaths because of cancer occurred. The most diagnosed cancer was female breast cancer with an yearly estimated incidence of 2.3 million. Additionally, the incidences of lung, colorectal, prostate, and stomach cancers were 11.4%, 10.0%, 7.3%, and 5.6%, respectively [2]. In response, research contentious work to discover new anticancer agents that have maximum effectiveness and minimal toxicity is still a vitaltrend in anticancer drug development and research [3].

Cancer cells are characterized by biochemical abnormalities [4]. Oxygen and nutrients are fundamental to cancer cells’ survival and proliferation; hence, cancer cells have to be located be near to blood vessels to have a high accessibility to the blood circulation [5]. Angiogenesis, the sprouting of new blood vessels from pre-existing vasculatures, is an essential factor in the process of cancer development and growth [6]. Current evidence confirmed that the primary tumor and the resulted metastasis depend on angiogenesis [7,8]. The angiogenesis process is under the control of various protein kinases, including the growth factors [7]. Amidst that growth factors, the vascular endothelial growth factor (VEGF) is of the most effective angiogenic determinants that can regulate angiogenesis and be engaged in the progression of a tumor [9,10].

VEGFs exert their angiogenic effects via binding with the different types of kinase domains of vascular endothelial growth factor receptors (VEGFRs 1, 2, and 3) [11]. VEGFR-2, the subtype receptor tyrosine kinase (RTK) responsible for angiogenesis [7], is expressed in various cancer cells and is responsible for the mediation of almost all the cellular responses to VEGF [12]. Thus, the VEGF/VEGFER-2 pathway is an competent target having vital selectivity for cancer cells [13]. The inhibition of the VEGFR-2/VEGF signaling pathway or the down-regulation of its response is a practical method for the discovery of new drugs for the treatment of several cancer types [12,14].

One of the hallmarks of cancer cells is their ability to avoid apoptosis, permitting unchecked and uncontrolled proliferation. Accordingly, the activation of apoptosis in cancer cells is a favorable strategy with which to defeat cancer [15]. The Bcl-2 family comprises members that are pro-apoptotic, such as Bax protein, which promotes cell apoptosis; and other members are antiapoptotic, such as Bcl-2 protein, which inhibits cell apoptosis [16]. The regulation of anti-apoptotic and pro-apoptotic proteins (Bax/Bcl-2 ratio) determines cell fate [16,17].

Several reports have detailed various small-molecule inhibitors of VEGFR-2 that target the ATP binding sites of the RTKs, resulting in diminished VEGF signal transduction [18,19,20,21,22,23,24]. These VEGFR-2 inhibitors can be broadly categorized into three types according to their modes of binding to the ATP binding pocket [14,18]. Type I inhibitors bind competitively to the adenine binding region of the ATP binding site in the active conformation (DFG-in) [18,25]. Type II inhibitors act on the ATP binding pocket and additionally function in the new allosteric hydrophobic pocket to stabilize the inactive conformation (DFG-out) [18]. Type II inhibitors provide better activity than type I by avoiding competitive inhibition [16]. Type III inhibitors adopt the inactive DFG-out conformation and bind covalently to the ATP binding site beyond the gatekeeper residue [25].

Over the last few years, a diverse range of VEGFR-2 inhibitors were designed, including sunitinib **I**, pazopanib **II,** sorafenib **III,** regorafenib **IV,** and trametinib **V** (Figure 1). However, drug resistance leads to decreased effectiveness and undesirable side effects. Therefore, it is essential to discover novel VEGFR-2 inhibitors for the treatment of cancer that have low toxicity and can overcome drug resistance [26].

### Rationale and Design

The study of the SAR of various VEGFR-2 inhibitors revealed that the core structures of most of them shared four fundamental features [16,26]: (i) A hetero aromatic ring (head) to occupy the catalytic ATP binding domain and participate in H-bonding interactions with the Cys919 amino acid residue the hinge region [18]. (ii) A “linker” segment occupying the gatekeeper residues that connects between the DFG domain and the ATP-binding domain. (iii) A hydrogen-bonding moiety (pharmacophore), such as urea or amide, interacts with the amino acids (Glu885 and Asp1046) via H-bonds in the DFG motif of the protein [4,15]. (iv) A terminal hydrophobic (tail) moiety directed toward the allosteric hydrophobic pocket, and binding with it through various hydrophobic interactions [15,16].

In continuation of our previous work on synthesizing new, potent, and safe anticancer agents [12,27,28,29,30,31,32], new nicotinamide derivatives were synthesized that keep the essential pharmacophoric features of VEGFR-2 inhibitors, aiming for the development of efficient anticancer agents that target VEGFR-2 inhuman cancer cell lines.

The designs of the new derivatives depend on the bioisosteric modifications at the four pharmacophoric features of the reported VEGFR-2 inhibitors. Firstly, it was reported that the nitrogen atom of the pyridine ring of sorafenib is a critical point for H-bonding interactions with Cys919 at the ATP binding domain. In addition, nicotinamidescaffold is required to occupy the hinge region (compound **VI**) [14]. Accordingly, nicotinamidemoiety was utilized as a head in the synthesized compounds to maintain the essential H-bonding interaction at the hinge region. In addition, it was reported that amide moiety may exert an anticancer effect [33]. With respect to the linker segment, it is usually 3–5 bond lengths [34]. The designed compounds involve a phenyl ring as a linker moiety to keep a tight fit in the gatekeeper area through hydrophobic interactions. Regarding the H-bonding pharmacophore moiety, we used a hydrazone moiety as an H-bonding center. This was based on the significant effect of the hydrazone moiety in the binding with VEGFR-2. Previously, our team developed compound **VII**, which has a promising VEGFR-2 inhibitory effect and comprises a hydrazone moiety as a pharmacophore group [35]. Finally, the terminal phenyl ring was kept to play its vital role and occupy the allosteric lipophilic pocket. The phenyl ring was substituted by diverse groups. Additionally, the hydrophobic moieties were modified to be amide moieties, in hopes of making extra binding interactions (Figure 2).

## 2. Results and Discussion

### 2.1. Chemistry

Figure 1 and Figure 2 demonstrate the synthetic pathways of the target compounds. Firstly, heating of nicotinic acid **1** with a thionyl chloride in 1,2 dichloroethane and DMF (catalytic amount) furnished nicotinoyl chloride **2** [16,28]. Reflux of nicotinoyl chloride **2** with 4-aminomethylbenzoate **3** in the presence of triethylamine in acetonitrile afforded the corresponding methyl ester derivative *N*-(4-acetylphenyl)nicotinamide **4** [16], which was treated with hydrazine hydrate to produce *N*-(4-(hydrazinecarbonyl)-phenyl)nicotinamide **5** [28]. The produced hydrazide derivative **5** was then reacted with appropriate aldehydes, namely, 2,4-dichlorobenzaldehyde, 3,4-dimethoxybenzaldehyde, and 3-(4-(dimethylamino)-phenyl)acrylaldehyde to afford the target compounds **6**–**8**, respectively (Figure 1).

Furthermore, nicotinoyl chloride **2** was further allowed to react with the commercially available *p*-aminoacetophenone **9** to yield the equivalent *N*-(4-acetylphenyl)nicotinamide **10** [27,28]. The latter was refluxed in ethanol and a catalytic amount of glacial acetic acid with appropriate amines, namely, thiosemicarbazide and 2,6-dinitrobenzohydrazide to give the target compounds **11** and **12**, respectively (Figure 2).

The structures of the synthesized compounds **6**, **7**, **8**, **11**, and **12** were authenticated by ^1^H-NMR, which showed the presence of two downfield singlet signals attributed to the NH protons. Taking compound **8** as a representative example, it showed two singlet signals at δ 11.57 and 10.71 ppm. The aromatic and olefinic protons appeared in the aromatic region from 9.14 to 6.73 ppm. The six protons of the two methyl groups appeared at the aliphatic area at 2.97 ppm as singlet signals.

### 2.2. Biological Testing

#### 2.2.1. In Vitro Anti-Proliferative Activities

Two human cancer cell lines (HCT-116; colorectal rectal cancer, and HepG2; hepatocellular carcinoma) were utilized to examine the anti-proliferative effects of the synthesized compounds. The VEGFR-2 protein is usually overexpressed in the HCT-116 and HepG-2 cell lines [36,37]. Accordingly, these two types of cancer cell lines were utilized in the presented work. MTT assay was employed with sorafenib as a standard anti-proliferative drug [35]. The results of cytotoxic activities (Table 1) revealed that compound **8** is the most active member showing strong cytotoxicity against HCT-116 and HepG2 with IC_50_ values of 5.4 and 7.1 µM, respectively. These results are comparable to sorafenib’s results (IC_50_ = 9.30 ± 0.201 and 7.40 ± 0.253 µM, respectively). Other members showed moderate activities with IC_50_ values ranging from 16.5 to 25.07 µM.

#### 2.2.2. In Vitro Vegfr-2 Enzyme Assay Inhibition

The obtained compounds were also investigated for their inhibitory potential against VEGFR-2 to examine the design and predict their inhibitory effect [16]. The VEGFR-2 inhibitory assay results (Table 1) indicated that compound **8** was the strongest inhibitor with an IC_50_ value of 77.02 nM (compare to sorafenib: IC_50_ = 53.65 nM). In addition, the other compounds showed moderate inhibitory effects with IC_50_ values ranging from 83.41 to 172.3 nM.

Depending on the outputs of cytotoxicity and VEGFR-2 kinase inhibition, compound **8**, was selected for further biological investigations.

##### Safety Pattern of the Tested Compounds

An in-vitro viability test was used to investigate the safety patterns of the tested compounds at different concentrations. The Vero cell lines were used as a non-cancerous model to investigate the safety of the targeted compounds. The results of the MTT assays indicate that the tested compounds have IC_50_ values ranging from 85.24 to 127.91 µM. Such values are very high in comparison to the corresponding values on the cancer cell lines, which reflect the high in vitro safety profiles of the tested members towards the examined non-cancerous cell lines (Table 2).

##### Selectivity Index (SI)

The selectivity index (SI) of a particular compound is the ratio of its toxic and effective concentrations (SI = IC_50_ against non-cancer cells/IC_50_ against cancer cells) [38]. Relatively, low SI (<1) means that the tested compound is toxic and cannot be used as a safe drug [39]. All the tested hybrids showed decreased potency against Vero cell lines. This finding encouraged us to investigate the selectivity profiles of the synthetized compounds. The selectivity index values of the synthesized compounds against cancer cells are indicated in Table 3. In general, all compounds showed SI of more than 4.2, indicating high selectivity of the tested compounds against cancer cell lines.

#### 2.2.3. Cell Cycle Analysis

Anticancer compounds exert their cytotoxicity by aborting the cellular growth at certain points. These points, phases, are distinguishable in the cell cycle, whose suppression causes the cessation of cell proliferation [15]. The ability of a compound to arrest the cell cycle is directly linked to apoptosis induction [40].

Flow cytometry analysis was utilized to investigate the efficacy of compound **8** on the cell cycle progression and apoptosis induction in the HCT-116 colon cancer cell line [41]. In the test, the HCT-116 cells were treated with compound **8** at a concentration of 5.4 µM, then incubated for 24 h. The cell cycle distribution was analyzed to identify the definite phase at which compound **8** can arrest the cell cycle. The results showed that compound **8** significantly declined the cell populations at G2/M and Pre-G1 phases, which were 12.91% (2.11-fold) and 3.05% (6.49-fold), compared to 27.29% and 19.82% for the control, respectively. Moreover, marked augmentation was observed in G0–G1 phase (55.62%; 1.42-fold) compared to the control (38.96%). This finding suggests that **8** arrested the cell cycle proliferation of the HCT-116 cell line at the G0–G1 phase (Table 4 and Figure 3).

#### 2.2.4. Induction of Apoptosis

As compound **8** produced arrest of HCT-116 cells at the G0–G1 phase, it is highly important to investigate its ability to induce apoptosis in the experienced cells. Its apoptosis induction ability was estimated using the Annexin V and PI double staining assay [42]. The results revealed significant elevations in cell percentages in both early (from 0.7 to 3.74%) and late apoptosis (from 1.73 to 14.23%) phases. These results indicate an increase in the total apoptosis from 3.05 to 19.82%—6.5-fold in comparison to the control. These findings clearly confirm that the cell death was attributable to functional apoptosis (Table 5 and Figure 4).

#### 2.2.5. Effects of Compound **8** on the Apoptotic Markers (Bax, Bcl-2, and caspase-8)

The effects of compound **8** on the levels of protein expression of the apoptotic markers (Bax, Bcl-2, and caspase-8) were estimated. HCT-116 cells were exposed to compound **8** at a concentration of 5.4 µM for 24 h; then the concentrations of the examined proteins were estimated using quantitative Real-Time Reverse-Transcriptase PCR (qRT-PCR). The results display high increases in caspase-8 levels (9.4-fold) and Bax (9.2-fold) compared to the control cells. Additionally, the Bcl-2 level was decreased by 3-fold comparing the control cells (Table 6). Such results indicate the significant apoptotic effect of the tested compound.

#### 2.2.6. Effects of Compound **8** on the Levels of the Immunomodulatory Proteins (TNF-α and IL-6)

Further investigations were conducted to assess the effects of compound **8** on the levels of immunomodulatory proteins (TNF-α and IL-6). HCT-116 cells were exposed to compound **8** at a concentration of 5.4 µM for 24 h, and then the concentrations of the examined proteins were determined using qRT-PCR. The results revealed a significant decrease in the level of TNF-α (92.37%) compared to the control (82.47%). On the other hand, compound **8** exerted a non-significant reduction in the level of IL-6. This suggests that compound **8** might induce apoptosis in HCT-116 cancer cells through immunomodulatory-dependent pathways (Table 4).

### 2.3. In Silico Studies

#### 2.3.1. Molecular Docking

A molecular docking experiment gives deep insight into the binding affinity between a specific compound and the target receptor [43]. The biological efficacy of a compound is linked to its better binding energy value and the degree of similarity in the binding mode compared the reference ligand [43]. To investigate the binding hallmarks of the targeted compounds against VEGFR-2, molecular docking studies were fulfilled by MOE. 2019.01 (Montreal, QC, Canada). The X-ray structure of VEGFR-2 co-crystallized with sorafenib was obtained from the Protein Data Bank (PDB ID: 4ASD) [14]. The docking process was firstly verified by re-docking of sorafenib inside the active pocket of VEGFR-2 (Figure 5). The low value (≤2.0 Å) of root mean square deviation (RMSD) of the docked conformer of sorafenib in the experimental crystal validates the utilized scoring function [27,44]. In the current study, the RMSD between the co-crystallized conformer and the re-docked one is 0.79 Å. This indicates the correctness of the employed docking protocol. The docking scores of the tested ligands are summarized in Table 7, and their binding features inside the active site of the target protein are depicted. The binding poses with the highest energy scores were selected for analysis. The outputted files from MOE software were visualized with Discovery Studio 4.0 software (San Diego, CA, USA).

Tyrosine kinase inhibitors have been divided into two types of binding patterns, (type I and II). Type I inhibitors bind to the kinase’s active form’s adenosine triphosphate (ATP) binding site, resulting in reduced selectivity, whereas type II inhibitors bind to the ATP site and the allosteric, hydrophobic site, resulting in great selectivity. This interaction takes place when the kinases are inactive. The kinase enzymes’ active and inactive phases are controlled by the conserved triad Asp–Phe–Gly (DGF). DGF-in conformation was observed in the active state, whereas DFG-out conformation was observed in the inactive state. The front and back pockets make up the VEGFR-2 active site’s basic architecture. A crucial residue related to the ATP binding front pocket is Cys919. Glu885 and Asp1046 are found in the back hydrophobic pocket. The Glu885 is located on the C helix, and Asp1046 is a crucial component of the triad [16].

A molecular docking study of sorafenib was carried out to study its binding interactions and orientation. This binding pattern showed a docking score of −36.23 kcal/mol with ten hydrogen bonds (H-bonds) and nine hydrophobic interactions. The carbonyl group of the *N-* methylpicolinamide head was involved in an H-bond with the vital amino acid Cys919 in the hinge region. Likewise, the pyridine ring was involved in three hydrophobic interactions against the hydrophobic pocket built by the amino acid, Leu1035, Phe918 (pi–pi stacking), and Ala866. Additionally, the spacer phenyl ring is bound to another hydrophobic pocket comprised of Lys 868, Val916, and Val899. Further, the urea moiety was involved in three H-bonds at the DFG motif, where the two NH made two H-bonds with Glu885 and the carbonyl group of the urea interacted with Asp1046. Finally, the terminal phenyl ring was involved in hydrophobic interactions (pi–pi stacking) with Leu1019, Ile1044, and His1026 (Figure 6).

All the newly synthesized compounds exhibited an interesting binding mode. Two representative compounds (**7** and **8**) were selected for analyzing their binding interactions and orientations against the active pocket of VEGFR-2.

Compound **7** demonstrated a promising binding pattern like that of sorafenib. This compound bonded tightly to the receptor with a binding energy value of −20.57 kcal/mol. In its binding, the pyridinyl group engaged with the hinge region forming a H-bond interaction with the vital amino acid Cys919. In addition, three hydrophobic interactions were displayed between this head and the hydrophobic pocket of (Leu840, Leu1035, and Ala866). The phenyl ring was stabilized in the gatekeeper district through four hydrophobic interactions with Val916, Val899, Lys868, and Cys1045. However, the pharmacophore hydrazone moiety achieved its required job by binding to the vital amino acids Glu885 and Asp1046 in the DGF motif. Moreover, the terminal dimethoxyphenyl hydrophobic tail completely fits in the allosteric site (Figure 7).

The best-scored pose (−22.71 kcal/mol) of compound **8** mimicked sorafenib’s key interactions. It kept the H-bonding interaction with the essential amino acid Cys919 in the hinge region via the nitrogen of the pyridine ring. The pyridine head was much more stabilized by many hydrophobic interactions generated with Leu840, Leu1035, Cys919, and Ala866. Moreover, the phenyl linker was enclosed in the gatekeeper area through four hydrophobic interactions with Val899, Val916, Lys868, and Cys1045. Furthermore, the hydrazone moiety interacted as an H-bond donor and acceptor, producing essential H-bonding interactions with Glu885 and Asp1046 at the DGF motif. Finally, the *N*,*N*-dimethyl aniline tail occupied the hydrophobic allosteric site (Figure 8).

#### 2.3.2. In Silico ADMET Analysis

The pharmacokinetic characteristics of the prepared compounds were analyzed by applying Discovery Studio 4.0 [45,46,47]. Sorafenib was utilized as a reference. The results of the ADME studies are listed in Table 8 and Figure 9. Except for compound **6,** all the investigated compounds demonstrated low or very low blood–brain barrier penetration. Compounds **7**, **8**, and **11** demonstrated good aqueous solubility; compounds **6** and **12** showed weak aqueous solubility. All synthesized compounds enjoyed high levels of absorption, whereas compound **13** showed a moderate one. All the tested members were anticipated to be CYP2D6 non-inhibitors. Finally, compounds **6**, **7**, **8**, and **12** can bind the plasma protein at a rate greater than 90%, and compound **11** was predicted to bind at a rate less than 90%.

#### 2.3.3. Lipinski’s Rule of Five and Veber’s Rule

The oral absorption of a certain compound is normally better if that compound satisfies at least three of the four Lipinski rules, listed below. (i) H-bond donors ≤ 5; (ii) H-bond acceptors ≤ 10; (iii) molecular weight < 500; (iv) logP < 5. The results revealed that all the synthesized compounds given in Table 9, and sorafenib, showed no violation of Lipinski’s rules.

Moreover, Veber’s rule was applied to the synthesized compounds. Veber’s rule depends on the molecular flexibility, as indicated by rotatable bonds number, and polar surface area to predict the oral bioavailability [49]. Compounds that have 10 or less rotatable bonds and a total polar surface area of 140 Å or less are predicted to have good oral bioavailability [49,50]. The results revealed that all the synthesized compounds obey Veber’s rule, except compound **12**, which has a total polar surface area of 158.01 Å.

#### 2.3.4. Toxicity Studies

Furtherly, the toxicity profile of the considered compounds was estimated using Discovery studio software. The results are demonstrated in Table 10. The results reveal that all compounds are non-mutagenic, except compounds **8** and **12**. Compounds **6**, **8**, **11**, and **12** showed carcinogenic potency TD_50_ values of 11.304, 29.149, 71.809, and 22.976 g/kg, respectively. These values are higher than that of sorafenib (14.244 g/kg). On the contrary, compound **7** showed lowers values of carcinogenic potency, TD_50_ (10.934 g/kg). Compounds **6**, **11**, and **12** showed maximum tolerated dose values of 0.125, 0.145, and 0.111 g/kg, respectively, which are higher than that of sorafenib (0.089 g/kg); compounds **7** (0.058 g/kg) and **8** (0.064 g/kg) were less tolerated than sorafenib. In addition, except for compound **11** (0.477 g/kg), the considered compounds showed oral LD_50_ values ranging from 1.282 to 2.143 g/kg, which are higher than that of sorafenib (0.823 g/kg). Additionally, the chronic lowest observed adverse effect level (LOAEL) was determined for all compounds. The results indicate LOAEL values ranging from 0.055 to 0.231 g/kg, higher than that of sorafenib (0.0048 g/kg). Finally, all the considered compounds were found to be non-irritant against the skin but to have mild irritant effects on the eye.

#### 2.3.5. MD Simulations

Molecular dynamics (MD) simulations are close to becoming routine in silico tools for drug design and discovery [51]. These studies have two major advantages. First is their explicit ability to examine any change that occurs in the ligand and the protein target, whether that change is structural or entropic. Second, MD studies compute the changes occurring at determined time steps over a very short period with atomic-level revolution [52]. Consequently, MD studies can correctly determine the kinetic and thermodynamic changes that occur through the process of ligand-protein binding [53]. These advantages present MD studies as a powerful tool to reveal the structure–function changes of the studied ligand–protein complex. They explore essential factors, such as ligand–target stability, in addition to ligand binding kinetics and energy [54].

The dynamic and conformational changes of backbone atoms of the VEGFR-2–compound **8** complex were evaluated by RMSD to explore the stability upon apo and 8 bonding states. Figure 10A reveals low RMSD values for VEGFR-2, compound **8**, and the VEGFR-2–compound **8** complex over 100 ns, except for a minor fluctuation after 60 ns. VEGFR-2 flexibility was inspected in terms of RMSF to investigate the fluctuated regions of the examined VEGFR-2 protein over the simulation. Figure 10B shows that compound 8 binding of VEGFR-2 does not make VEGFR-2 highly flexible. The radius of gyration (Rg) of the examined VEGFR-2 enzyme was explored. As shown in Figure 10C, a slight degree of fluctuation of the VEGFR-2 enzyme throughout 100 ns indicates the compactness of the VEGFR-2 -compound **8** complex. Additionally, solvent accessible surface area (SASA) was investigated over 100 ns to investigate the VEGFR-2–compound **8** complex’s interaction with the surrounding solvents. Amazingly, the VEGFR-2 enzyme featured a reduced SASA value at 100 ns compared to 0 ns (Figure 10D), indicating the stability of the VEGFR-2–compound **8** complex. Finally, H-bonding in the VEGFR-2–compound **8** complex was revealed to be up to three H-bonds (Figure 10E).

#### 2.3.6. MM-PBSA Studies

The molecular mechanical energies with the Poisson–Boltzmann Born and surface area continuum solvation (MM-PBSA) method is an accurate approach to computing the exact free energy of the binding of an examined compound inside a specific protein [55]. The exact binding energy of the VEGFR-2–compound **8** complex in the final 20 ns of the MD was found with an interval of 100 ps from the resulted MD trajectories. Compound **8** demonstrated binding free energy of 220 KJ/mol with the VEGFR-2 enzyme (Figure 11A). Furthermore, the contribution of every amino acid of VEGFR-2 in the obtained binding free energy after the interaction with compound **8** was analyzed. The achieved findings gave an insight into the fundamental residues that had a pivotal role in the binding of the VEGFR-2–compound **8** complex. It was found that ASP-852, ASP-857, GLU-917, GLU-934, and GLU-1038 residues of the VEGFR-2 contributed to the binding energy by more than −20 KJ/mol (Figure 11B).

#### 2.3.7. Flexible Alignment

3D flexible alignment of compound **8** with sorafenib was studied. The result of flexible alignment revealed the general good overlap of compound **8** with sorafenib with the same spatial orientation. In detail, nicotinamide, phenyl, hydrazone, and 3-(4-(dimethylamino)phenyl)allylidene moieties of compound **8** showed the same orientation of the *N*-methylpicolinamide, phenoxy, urea, and 4-chloro-3-(trifluoromethyl)phenyl) moieties as sorafenib, respectively (Figure 12).

## 3. Conclusions

Five new nicotinamide derivatives were designed and synthesized as anti-proliferative VEGFR-2 inhibitors. The synthesized compounds showed promising antiproliferative activities against (HCT-116 and HepG2) cell lines, and good VEGFR-2 inhibitory activities. The most active member, **8**, exhibited IC_50_ values of 5.4 and 7.1 µM against HCT-116 and HepG2, respectively, and an IC_50_ value of 77.02 nM against VEGFR-2. Deep biological studies were conducted for compound **8**. It showed a promising apoptotic ability (6.5-fold in comparison to the control) to arrest the cell cycle at the G0–G1 phase. In addition, compound **8** produced significant increases in the expression levels of Bax (9.2-fold) and caspase-8 (9.4-fold) compared to the control cells. Furthermore, it caused a significant reduction in the anti-apoptotic factor (Bcl-2) by 3-fold compared to the control cells. Furthermore, it caused a significant reduction in the level of TNF-α (92.37%) compared to the control (82.47%), and a non-significant reduction in the level of IL-6. Docking studies explained the binding modes of the considered compounds. They showed binding modes like that ofsorafenib. In silico ADMET and toxicity studies indicated the low toxicity of the considered compounds, and a good range of pharmacokinetic properties. The molecular dynamics simulations revealed the high stability of the most active member, **8**, in the active site of VEGFR−2. Compound **8** is considered a good lead for further chemical modifications and deep biological testing.

## 4. Experimental

### 4.1. Chemistry

#### 4.1.1. General

All reagents, chemicals, and apparatus are shown in the Appendix A. The solvents and chemicals were purchased from El-Gomhouria Co. For Trading Drugs, Chemicals and Medical Supplies, Cairo, Egypt. Compounds **2**, **3**, **4**, **5**, and **10** were previously reported [56,57].

#### 4.1.2. Synthesis of Compounds **6**, **7**, and **8**

A mixture of hydrazide derivative **5** (0.256 g, 0.001 mol) and 0.001 mol of the suitable aromatic aldehyde (0.001 mol), namely, 2,4-dichlorobenzaldehyde, 3,4-dimethoxybenzaldehyde, or 4-(dimethylamino)cinnamaldehyde, were refluxed in 30 mL of ethanol (absolute) containing glacial acetic acid (0.15 mL) for 2 h. After reaction accomplishment, the reaction mixture was cooled, filtered, dried, and recrystallized from ethanol to afford compounds **6**, **7**, and **8**, respectively.

##### (*E*)-*N*-(4-(2-(2,4-Dichlorobenzylidene)hydrazine-1-carbonyl)phenyl)-nicotinamide (**6**)

White crystal (yield, 71%); m.p. = 278–280 °C; C_20_H_14_C_l2_N_4_O_2_ (413.2580 g/mol); IR (KBr) *ν* cm^−1^: 3301, 3228 (NH), 3034 (CH aromatic), 1644 (C=O), 1593 (C=N); ^1^H-NMR (400 MHz, DMSO-*d*_6_) δ 12.08 (s, 1H), 10.73 (s, 1H), 9.14 (d, *J* = 2.3 Hz, 1H), 8.92–8.68 (m, 2H), 8.33 (dt, *J* = 7.9, 2.0 Hz, 1H), 8.05–8.03 (d, *J* = 8 Hz, 1H), 8.00–7.98 (d, *J* = 8 Hz 2H), 7.96–7.94(d, *J* = 8 Hz, 2H), 7.70 (d, *J* = 2.2 Hz, 1H), 7.59 (dd, *J* = 8.0, 4.8 Hz, 1H), 7.52 (dd, *J* = 8.6, 2.2 Hz, 1H); ^13^C-NMR (101 MHz, DMSO-*d*_6_) δ 164.92, 163.13, 152.78, 149.18, 142.84, 142.66, 136.06, 135.50, 134.29(2C), 131.22, 130.77(2C), 129.81, 129.07(2C), 128.54, 128.46, 124.03, 120.10.

##### (*E*)-*N*-(4-(2-(3,4-Dimethoxybenzylidene)hydrazine-1-carbonyl)phenyl)-nicotinamide (**7**)

Off-white crystal (yield, 74%); m.p. = 250–252 °C; C_22_H_20_N_4_O4 (404.4260 g/mol); IR (KBr) *ν* cm^−1^: 3311, 3248 (NH), 3037 (CH aromatic), 2945 (CH aliphatic), 1644 (C=O), 1598 (C=N); ^1^H-NMR (400 MHz, DMSO-d6) δ 11.75 (s, 1H), 10.75 (s, 1H), 9.16 (d, *J* = 2.2 Hz, 1H), 8.80 (dd, *J* = 4.8, 1.6 Hz, 1H), 8.52–8.27 (m, 2H), 7.99–7.97 (d, *J* = 8 Hz, 2H), 7.96–7.94 (d, *J* = 8 Hz, 2H), 7.60 (dd, *J* = 7.9, 4.8 Hz, 1H), 7.37 (s, 1H), 7.22 (d, *J* = 8.2 Hz, 1H), 7.04 (d, *J* = 8.3 Hz, 1H), 3.82 (s, 6H); ^13^C-NMR (101 MHz, DMSO-d6) δ 164.89, 162.91, 152.80, 151.20, 149.55, 149.24, 148.26, 142.39, 136.06, 130.82(2C), 129.04, 128.93, 127.57, 124.02, 122.38, 120.08, 111.95, 108.65, 56.04, 55.92.

##### *N*-(4-(2-((*1E*,*2E*)-3-(4-(Dimethylamino)phenyl)allylidene)hydrazine-1-carbonyl) phenyl)nicotinamide (**8**)

White crystal (yield, 67%); m.p. = 260–262 °C; C_24_H_23_N_5_O_2_ (413.4810g/mol); IR (KBr) *ν* cm^−1^: 3308, 3239 (NH), 3033 (CH aromatic), 2895 (CH aliphatic), 1643 (C=O), 1608 (C=N); ^1^H-NMR (400 MHz, DMSO-*d*_6_) δ 11.57 (s, 1H), 10.71 (s, 1H), 9.14 (s, 1H), 8.80 (d, *J* = 4.7 Hz, 1H), 8.33 (d, *J* = 8.0 Hz, 1H), 8.19 (d, *J* = 9.1 Hz, 1H), 7.94 (s, 4H), 7.61 (t, *J* = 6.5 Hz, 1H), 7.47 (d, *J* = 8.3 Hz, 2H), 6.92 (d, *J* = 16.0 Hz, 1H), 6.83 (d, *J* = 9.3 Hz, 1H), 6.73 (d, *J* = 8.4 Hz, 2H), 2.97 (s, 6H); ^13^C-NMR (101 MHz, DMSO-d6) δ 164.89, 162.58, 152.82, 151.16, 150.89, 149.23, 144.29, 142.29, 140.19, 138.74, 136.08, 133.38, 130.85, 129.09(2C), 128.89(2C), 124.13, 124.05, 121.05, 120.01, 112.49, 38.79.

#### 4.1.3. Synthesis of Compounds **11** and **12**

To a solution of compound **10** (0.24 g, 0.001 mol) in absolute ethanol (20 mL) and 0.15 mL of glacial acetic acid, the appropriate amine derivative, namely, thiosemicarbazide or 2,5-dinitrobenzohydrazide (10 mmol), was added. The reaction mixture was refluxed for 8 h. The obtained solids were filtered, washed dried, and recrystallized from ethanol to give the target compounds **11** and **12**.

##### *N*-(4-(1-(2-Carbamothioylhydrazono)ethyl)phenyl)nicotinamide (**11**)

Brownish white crystal (yield, 61%); m.p. = 239–241 °C; C_15_H_15_N_5_OS (313.3790 g/mol); IR (KBr) *ν* cm^−1^: 3346, 3261 (NH), 3171 (CH aromatic), 1659 (C=O), 1594 (C=N); ^1^H-NMR (400 MHz, DMSO-d6) δ 10.56 (s, 1H), 10.18 (s, 1H), 9.13 (d, *J* = 2.8 Hz, 1H), 8.78 (dd, *J* = 4.8, 1.7 Hz, 1H), 8.33–8.30 (m, 1H), 8.00–7.93 (m, 3H), 7.84–7.79 (m, 2H), 7.71–7.62 (m, 1H), 7.59 (dd, *J* = 8.0, 4.7 Hz, 1H), 2.30 (s, 3H); ^13^C-NMR (101 MHz, DMSO-d6) δ 179.19, 170.39, 164.62, 152.67, 149.18, 147.86, 140.30, 136.01, 133.47, 131.76, 130.96, 127.64, 124.01, 120.10, 14.23. This compound has *Z* and *E* forms at 1:0.5

##### (*E*)-*N*-(4-(1-(2-(2,6-Dinitrophenyl)hydrazono)ethyl)phenyl)nicotinamide (**12**)

Reddish crystal (yield, 61%); m.p. = 265–267 °C; C_21_H_16_N_6_O_6_ (420.3850g/mol); IR (KBr) *ν* cm^−1^: 3299 (NH), 3095 (CH aromatic), 2927 (CH aliphatic) 1679 (C=O), 1611 (C=N);^1^H-NMR (400 MHz, DMSO-*d*_6_) δ 11.15 (s, 1H), 10.67 (s, 1H), 9.14 (s, 1H), 8.92 (s, 1H), 8.80 (s, 1H), 8.43 (d, *J* = 9.5 Hz, 1H), 8.33 (s, 1H), 8.15 (d, *J* = 9.6 Hz, 1H), 8.01 (d, *J* = 7.7 Hz, 2H), 7.92 (d, *J* = 8.8 Hz, 2H), 7.61 (s, 1H), 2.48 (s, 3H); ^13^C-NMR (101 MHz, DMSO-d6) δ 164.77, 153.39, 152.76, 149.22, 144.93, 142.92, 137.69, 136.04, 132.80, 130.63, 130.23, 127.73(2C), 127.70, 124.03, 123.56, 120.80(2C), 120.41, 117.09, 13.99.

### 4.2. Biological Testing

#### 4.2.1. In Vitro Anti-Proliferative Activity

MTT procedure [58,59] was applied as represented in the Appendix A.

#### 4.2.2. In Vitro VEGFR-2 Enzyme Inhibition Assay

VEGFR-2 inhibitory activity was explored using the Human VEGFR-2 ELISA kit [60], as represented in Appendix A.

#### 4.2.3. Flow Cytometry Analysis for Cell Cycle

This was explored utilizing propidium iodide (PI) staining and flow cytometry analysis for compound **8,** as represented in Appendix A [61].

#### 4.2.4. Flow Cytometry Analysis for Apoptosis

Flow cytometry cell apoptosis analysis was applied for compound **8**, as represented in Appendix A [62,63,64].

#### 4.2.5. Quantitative Real-Time Reverse-Transcriptase PCR (qRT-PCR) Technique

The effects of compound **8** on the expression of cleaved caspase-3, Bax, Bcl-2, TNF-α, and IL-6 were determined using qRT-PCR as represented in Appendix A [65,66,67].

### 4.3. In Silico Studies

#### 4.3.1. Docking Studies

The synthesized compounds were docked against the crystal structure of VEGFR-2 [PDB: 4ASD, resolution: 2.03 Å] using MOE2019.01 software as represented in Appendix A [24].

#### 4.3.2. ADMET Studies

ADMET descriptors were determined using Discovery studio 4.0 [23] as represented in (Appendix A).

#### 4.3.3. Toxicity Studies

The toxicity parameters were calculated using Discovery studio 4.0 [22], as represented in Appendix A.

##### Molecular Dynamics Simulation & MM/PBSA

MD simulations and MM/PBSA (molecular mechanics/Poisson Boltzmann surface Area) were performed using GROMACS [68] as represented in the Appendix A.

## Data Availability

Not applicable.

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
