# Peer review of "Design, Synthesis, In Silico and In Vitro Studies of New Immunomodulatory Anticancer Nicotinamide Derivatives Targeting VEGFR-2"

_molecules, 2022, doi:10.3390/molecules27134079_

Round 1

Reviewer 1 Report

The work on the “Design, synthesis, in silico and in vitro studies of new im- 2 munomodulatory anticancer nicotinamide derivatives targeting 3 VEGFR-2” is a valuable well-done interesting contribution.

An extensive English revision is needed and typo errors were observed in the whole manuscript.

I would like to recommend the authors to re-write a lot of sentences due to the overlapping with a lot of other articles, the sentences were copied as its, and the similarity rate was too high. The following lines should be edited accordingly: 48, 57-61, 71-73, 78-79, 85-86, 100-110, 128, 146-147, 193-194, 206-210, 265-270, and 290-293.

  • Abstract
  • Line 32 control which controls positive or negative??
  • Can you add a sentence regarding the chemical characterization methods of these compounds “ NMR, HRMS, IR “?? To abstract.
  • Line 39 in silicoADMET adds space between the two words
  • Add a conclusion to your abstract
  • Introduction
  • Edit the lines mentioned before due to the overlapping
  • I would like to a statistical data regarding cancer to the introduction from recent publications.
  • The introduction is too long it should be summarized more
  • It is clear that the authors are strongly continuing their VEGFR-2 inhibitors discovery story, but this is not mean that they should make high self-citations, you can cite rather references with amid or methoxyphenyl derivatives with anticancer activities like https://doi.org/10.1155/2021/6633297 to reduce your self-citation
  • Results and discussion
  • The chemistry section was well written
  • Edit the lines mentioned before due to the overlapping
  • Line 188, these edit as These
  • Line 189 add space between and7.40 ±
  • Use IC50 with subscript in the whole manuscript (edit line 190)
  • Add respectively to the last line 191
  • Write HepG2 in the whole MS same as.
  • Can you add what was the control to the table 2
  • In figure 3 you have to add the error bars
  • Regarding the induction of apoptosis method can you add the following ref. for annexin V https://doi.org/10.1080/01480545.2021.1935397.
  • You can improve the resolution of figure 4
  • The molecular docking studies were well presented and discussed
  • Regarding the ADMET analysis recent publications are recommended to be cited.
  • Improve the resolution of figure 9
  • Experimental
  • Can you mention from where all chemicals and reagents were ordered?
  • What did you mean with C20H14Cl2N4O2 (413.26) this value is the molecular weight, if yes, how did you find it? Did you use HRMS? Add g/mol if MWt.
  • Line 463 deletes the (2C) cuz there are two signals that come with each other for the methoxy groups they could not be three Carbone atoms there.
  • Regarding compound 8 the most important one, I could not find the Carbone spectrum in the supp. file?
  • Regarding the H-NMR of compound 11 the spectrums is not pure why ?? and compound 12 you don’t have 13C NMR ? and the baseline of H-NMR was very bad why, can you do that again?
  • Can you use a recent publication regarding the flow cytometry analysis, newer than 58 and 59 ref.s.
  • The Conclusion was well written
  • Control again all references as the journal style  

 Best wishes

Author Response

Reviewer 1

The work on the “Design, synthesis, in silico and in vitro studies of new im- 2 munomodulatory anticancer nicotinamide derivatives targeting 3 VEGFR-2” is a valuable well-done interesting contribution.

Thank you for your efforts and your valuable comments. All comments were considered in high interest and all changes were highlighted in the revised manuscript.

  • An extensive English revision is needed, and typo errors were observed in the whole manuscript.

Response: Done

  • I would like to recommend the authors to re-write a lot of sentences due to the overlapping with a lot of other articles, the sentences were copied as its, and the similarity rate was too high. The following lines should be edited accordingly: 48, 57-61, 71-73, 78-79, 85-86, 100-110, 128, 146-147, 193-194, 206-210, 265-270, and 290-293.

Response: All these lines were rephrased. In addition, the plagiarism was checked by iThenticate

  • Abstract
  • Line 32 control which controls positive or negative??
  • Can you add a sentence regarding the chemical characterization methods of these compounds “NMR, HRMS, IR “?? To abstract.
  • Line 39 in silicoADMET adds space between the two words
  • Add a conclusion to your abstract.

Response:  All these valuable comments were considered and adjusted in the revised manuscript.

  • Introduction
  • Edit the lines mentioned before due to the overlapping
  • I would like to a statistical data regarding cancer to the introduction from recent publications.
  • The introduction is too long it should be summarized more
  • It is clear that the authors are strongly continuing their VEGFR-2 inhibitors discovery story, but this is not mean that they should make high self-citations, you can cite rather references with amid or methoxyphenyl derivatives with anticancer activities like https://doi.org/10.1155/2021/6633297 to reduce your self-citation.

Response: All these valuable comments were considered and adjusted in the revised manuscript.

  • Results and discussion
  • The chemistry section was well written
  • Edit the lines mentioned before due to the overlapping
  • Line 188, these edit as These
  • Line 189 add space between and7.40 ±
  • Use IC50with subscript in the whole manuscript (edit line 190)
  • Add respectively to the last line 191
  • Write HepG2 in the whole MS same as.
  • Can you add what was the control to the table 2
  • In figure 3 you have to add the error bars
  • Regarding the induction of apoptosis method can you add the following ref. for annexin V https://doi.org/10.1080/01480545.2021.1935397.
  • You can improve the resolution of figure 4
  • The molecular docking studies were well presented and discussed
  • Regarding the ADMET analysis recent publications are recommended to be cited.
  • Improve the resolution of figure 9.

Response: Thank you for these comments. All these valuable comments were considered and adjusted in the revised manuscript.

  • Experimental
  • Can you mention from where all chemicals and reagents were ordered?

Response: Done

  • What did you mean with C20H14Cl2N4O2 (413.26) this value is the molecular weight, if yes, how did you find it? Did you use HRMS? Add g/mol if MWt.

Response: this is the theoretical molecular weight. We cannot carry out HRMS because it is not available in our country.

  • Line 463 deletes the (2C) cuz there are two signals that come with each other for the methoxy groups they could not be three Carbone atoms there.

Response: done

  • Regarding compound 8 the most important one, I could not find the Carbone spectrum in the supp. file?

Response: Carbone spectrum was added into the supp data.

  • Regarding the H-NMR of compound 11 the spectrums is not pure why ?? and compound 12 you don’t have 13C NMR ? and the baseline of H-NMR was very bad why, can you do that again?

Response: Compound 11 has solvent impurities. 13C NMR of Compound 12 has been carried out in the revised manuscript.

  • Can you use a recent publication regarding the flow cytometry analysis, newer than 58 and 59 ref.s.

Response: done

  • The Conclusion was well written.

Response: thank you

  • Control again all references as the journal style.

Response: done

Reviewer 2 Report

The article "Design, synthesis, in silico and in vitro studies of new immunomodulatory anticancer nicotinamide derivatives targeting VEGFR-2" by Yousef et al. describes the design and synthesis of five novel nicotinamide derivatives as VEGFR-2 inhibitors as well as their antiproliferative activity and VEGFR-2 inhibition in vitro. The authors also performed cell cycle analysis and investigated the influence on apoptosis. In silico docking, molecular dynamics simulation, and ADMET studies were also conducted.

The authors describe the SAR of known VEGFR-2 inhibitors and the design of novel potential inhibitors. The introduction is thorough and very well written, including relevant and up-to-date references. There are a couple of errors in the introduction - e.g. lines 76 and 77 - Bax protein is mentioned to be pro- and anti-apoptotic. However, I believe the authors wanted to say that the Bcl-2 family comprises both pro-apoptotic members, such as Bax protein, and anti-apoptotic ones, such as Bcl-2 protein.

Concerning the chemistry part, the synthetic routes employed are straightforward and simple, with good yields. The enumeration of compounds in Fig. 2. should be changed in order to be in line with the text, schemes and Table 1. (i.e. 7a, 7b, and 7c should be 6, 7, and 8). The number of synthesized compounds is rather low - only 5 new compounds were synthesized and tested. Therefore, SAR could not be commented on. The number of tested cancer cell lines is also too low - the authors chose only two cancer cell lines (HepG2 and HCT116). Therefore, the selectivity could not be discussed as well. The authors emphasize that they aimed at synthesizing less toxic anticancer agents. However, they only performed in silico toxicity studies. The authors should have performed an antiproliferative assay on normal cell lines (for example, human fibroblasts, or, if not available, at least immortalized Hek293) and discussed selectivity and cytotoxicity. In the "Biological testing" part, the authors state that compound 8 showed a stronger antiproliferative effect against the tested cell lines than the standard drug sorafenib. However, the IC50 value for HepG2 is comparable with the IC50 of sorafenib (7.1 uM vs. 7.4 uM). Also, they can not say that the values were exceeding that of sorafenib as they are actually lower. Also, HCT116 is misspelt as HCT16 throughout the text.

In the Supplementary Materials, the authors describe the in vitro antiproliferative assay procedure and say they treated the cells with test compounds at 4 different concentrations (0.1, 10, 100 and 1000 uM). However, the concentrations of 100 and 1000 uM are not relevant as the IC50 values are under 100 uM, to be more specific - even under 50 uM. The concentration range should have been adjusted accordingly (e.g. 1, 5, 10, 25, 50 uM). The authors also do not mention whether they performed this experiment at least in duplicate or triplicate for each conc.

Furthermore, concerning the cell cycle analysis, what is the rationale for treating HCT116 with compound 8 at 7.1 uM concentration? The usual concentrations used are IC50 plus one higher concentration (usually 2 x IC50). The chosen concentration is the IC50 for HepG2 and not HCT116. To clarify, this concentration can be used (the results would probably be comparable), but it seems to be the authors have confused the two IC50 values. Furthermore, and this concerns the whole manuscript, the lack of statistics - when discussing that something is significantly different, the p-value should be defined prior, and the exact p-value stated (lines 215-219; 230-233). The two graphs in Fig. 3. above the Cell Cycle Distribution histogram should either be deleted or explained. Additionally, the legend Cell Cycle Distribution histogram is incorrect.

In the "In silico ADMET analysis" part, line 338 is not in line with Table 6. According to Table 6., tested compounds are not CYP2D6 inhibitors, while the text says otherwise. Furthermore, it is not clear what Fig. 9. means. Additionally, according to Table 7., both compounds 12 and 8 are predicted to be mutagens. However, the text only mentions 12 as a mutagen.

Finally, what concerns me the most is that the authors cited their irrelevant articles in the experimental part (ref. 65-76). For example, references 65-68 are cited in Docking studies. The cited articles containing docking and MD studies for completely different proteins. I am sure there are other articles on VEGFR-2 docking that are far more appropriate. Also, the articles cited in ADMET studies (ref. 69-71) are about potential SARS-CoV2 and are unrelated to this topic. The same goes for self-citations in the Toxicity and MD sections. It is unethical and completely unacceptable to self-cite irrelevant articles to boost one's citations. To sum it all up, a major revision is necessary.

Author Response

Reviewer 2

The article "Design, synthesis, in silico and in vitro studies of new immunomodulatory anticancer nicotinamide derivatives targeting VEGFR-2" by Yousef et al. describes the design and synthesis of five novel nicotinamide derivatives as VEGFR-2 inhibitors as well as their antiproliferative activity and VEGFR-2 inhibition in vitro. The authors also performed cell cycle analysis and investigated the influence on apoptosis. In silico docking, molecular dynamics simulation, and ADMET studies were also conducted.

Thank you for your efforts and your valuable comments. All comments were considered in high interest and all changes were highlighted in the revised manuscript.

  • The authors describe the SAR of known VEGFR-2 inhibitors and the design of novel potential inhibitors. The introduction is thorough and very well written, including relevant and up-to-date references. There are a couple of errors in the introduction - g. lines 76 and 77 - Bax protein is mentioned to be pro- and anti-apoptotic. However, I believe the authors wanted to say that the Bcl-2 family comprises both pro-apoptotic members, such as Bax protein, and anti-apoptotic ones, such as Bcl-2 protein.

Response: Thank you for this notice. The sentence was corrected.

  • Concerning the chemistry part, the synthetic routes employed are straightforward and simple, with good yields. The enumeration of compounds in Fig. 2. should be changed in order to be in line with the text, schemes and Table 1. (e.7a7b, and 7c should be 67, and 8).

Response: corrected

  • The number of synthesized compounds is rather low - only 5 new compounds were synthesized and tested. Therefore, SAR could not be commented on.

Response: Response: Thank you for this notice. There are some points.

  • The presented work relied on a well-structured design of the pharmacophoric features of VEGFR-2 inhibitors. Accordingly, we didn't synthesize a large library of compounds for screening. This kind of work was a goal for our team as it saves time, effort, costs and protects the environment from the drawbacks of chemical synthesis and the use of excessive organic solvents.
  • During the design stage, we were interested in some considerable key points
  • We selected nicotinamide as heterocyclic systems to occupy the hinge region of the active site.
  • We used different terminal hydrophobic moieties

Accordingly, we have reached valuable results regarding SAR as shown in the manuscript.

  • Synthesis of new compounds at this time is very difficult and takes very long time and the available time for revision is not too long to do that.
  • The manuscript did not stop at chemical synthesis. We carried out deep biological studies and many in silico investigations including (docking, ADMET, toxicity, and MD simulation).

  • The number of tested cancer cell lines is also too low - the authors chose only two cancer cell lines (HepG2 and HCT116). Therefore, the selectivity could not be discussed as well. The authors emphasize that they aimed at synthesizing less toxic anticancer agents. However, they only performed in silicotoxicity studies. The authors should have performed an antiproliferative assay on normal cell lines (for example, human fibroblasts, or, if not available, at least immortalized Hek293) and discussed selectivity and cytotoxicity.

Response:

  • We used two cell lines that are characterized by an overexpression of VEGFR-2. So that, we went directly towards the biological target.
  • For the cytotoxicity against normal cells, it has been carried out in the revised manuscript.
  • In the "Biological testing" part, the authors state that compound 8showed a stronger antiproliferative effect against the tested cell lines than the standard drug sorafenib. However, the IC50 value for HepG2 is comparable with the IC50 of sorafenib (7.1 uM  7.4 uM). Also, they can not say that the values were exceeding that of sorafenib as they are actually lower. Also, HCT116 is misspelt as HCT16 throughout the text.

Response: All these issues were adjusted

  • In the Supplementary Materials, the authors describe the in vitroantiproliferative assay procedure and say they treated the cells with test compounds at 4 different concentrations (0.1, 10, 100 and 1000 uM). However, the concentrations of 100 and 1000 uM are not relevant as the IC50 values are under 100 uM, to be more specific - even under 50 uM. The concentration range should have been adjusted accordingly (g. 1, 5, 10, 25, 50 uM). The authors also do not mention whether they performed this experiment at least in duplicate or triplicate for each conc.

Response: thank you for this notice. The values of concentrations were corrected in supplementary data. In addition, all experiments were carried out three times at least.

  • Furthermore, concerning the cell cycle analysis, what is the rationale for treating HCT116 with compound 8at 7.1 uM concentration? The usual concentrations used are IC50 plus one higher concentration (usually 2 x IC50). The chosen concentration is the IC50 for HepG2 and not HCT116. To clarify, this concentration can be used (the results would probably be comparable), but it seems to be the authors have confused the two IC50 Furthermore, and this concerns the whole manuscript, the lack of statistics - when discussing that something is significantly different, the p-value should be defined prior, and the exact p-value stated (lines 215-219; 230-233). The two graphs in Fig. 3. above the Cell Cycle Distribution histogram should either be deleted or explained. Additionally, the legend Cell Cycle Distribution histogram is incorrect.

Response: we apologize for this typo. Actually, we used the IC50 concentration of compound 8 against HCT-116 as described in supplementary data. This typo was corrected.  All recommendations were carried out

  • In the "In silico ADMET analysis" part, line 338 is not in line with Table 6. According to Table 6., tested compounds are not CYP2D6 inhibitors, while the text says otherwise. Furthermore, it is not clear what Fig. 9. means. Additionally, according to Table 7., both compounds 12and 8 are predicted to be mutagens. However, the text only mentions 12 as a mutagen.

Response: thank you for these notices. All these issues were corrected. Additional explanation fot figure 9 was added.

  • Finally, what concerns me the most is that the authors cited their irrelevant articles in the experimental part (ref. 65-76). For example, references 65-68 are cited in Docking studies. The cited articles containing docking and MD studies for completely different proteins. I am sure there are other articles on VEGFR-2 docking that are far more appropriate. Also, the articles cited in ADMET studies (ref. 69-71) are about potential SARS-CoV2 and are unrelated to this topic. The same goes for self-citations in the Toxicity and MD sections. It is unethical and completely unacceptable to self-cite irrelevant articles to boost one's citations. To sum it all up, a major revision is necessary.

Response: New updated and relevant refences were cited in the docking, ADMET, toxicity, and MD studies

Reviewer 3 Report

In this article the authors discuss the design, synthesis and evaluation of five nicotinamide derivatives. Also, they made some in silico studies such as, docking, ADMET parameters and molecular dynamics.

I have some comments about the paper:

-Why were the cellular lines HCT-116 and HepG-2 chosen for the assays? Have these cellular lines specially overexpressed VEGRF-2?

-The schemes about the synthetic routes (schemes 1 and 2) aren’t detailed enough. Some conditions (temperature, time of reaction, …) and yields are missing. Also, ethanol is EtOH not EthOH.

-If in the design proposed, there is a hydrophobic tail, why didn’t the authors use an aliphatic aldehyde? Also, it is also interesting to use and aldehyde with an CF3 group (due to its pharmaceutical properties and to have some similarity with sorafenib).

-In table 1, the SEM deviations of the IC50 in the VEGFR-2 are missing. Also, it should state the number of times the experiments were performed (In the Supplementary data only says that the experiments were performed in triplicate for the PCR).

-The explanation of figures captions and tables should be more elaborated.

-Concerning the molecular docking:

The versions of software used are from several years ago (MOE.14,…) why haven’t authors used more actual versions which have been improved?.

If they are using MOE for the docking, why have authors used other visualizers if it could be done with MOE? Also, some figures (2D image of figure 5) were obtained with MOE and others with other visualizers, is better if all the figures are obtained with the same software.

The molecules in figures 5, 6, 7 and 8 have different orientations. They should all have the same orientations in order to compare them and to easily locate  the residues and interactions.

In the article the PDB used is 4ASD, but in the Supplementary data there are two PDB listed. Why was the PDB:4ASD selected?

In figure 5: Of the two molecules present in the figure (red and green) it should be address which is the re-docked conformer and which one is the co-crystalized conformer. Also, in the 2D image there is no alienation of the two molecules (as it is in the 3D image).

When the protein was being prepared for the docking, authors removed all the water molecules. But, there is a water molecule (H2O 2090) that has interactions with Leu840 and with a carbonyl group from sorafenib.

The binding free energies (ΔG) are referred as docking score, but the “S score” should be also given.

Amino acids Cys919, Glu885 and Asp1046 are referred to as essential and key amino acids. The reason of this should be discussed.

Some residues are different in the text than in the figures. For example, in line 306 it says “Cys1945” but in figure 6 it appears “Cys1045”. In line 321 says “Glu883” but in figure 7 it appears “Glu885”, in line 322 says “Asp1044” but in figure 7 it appears “Asp1046”.

Also, the hydrophobic interactions in residues that don’t have aromatic rings and thus that don’t form p-p stacking aren’t represented.

In figures 6, 7 and 8 there are three images in each (surface, 3D and 2D binding modes) but they are repetitive and show nothing different. With one image that show the interactions is enough.

One figure that could be interesting is a superposition of sorafenib and compound 8.

In line 298 it says “(…) the most cytotoxic derivatives 7 and 8 were selected for analyzing their binding 298 interactions and orientations.” But compound 7 is the third most active with respect to VEGFR-2 (with an IC50 of 112.6 nM, compound 11 is the second with an IC50 of 83.41 nM) and the forth respect the two cellular lines (compounds 8, 6 and 12 are more active).

-With respect to the in silico ADMET properties:

Why are the parameters calculated with other software (Discovery Studio 4.0)? It could be done with MOE.

If the oral distribution is going to be discussed (table 7), the rule of five of Lipinsky could be addressed and if the compounds have any violations of these rules.

In line 352 says: “The results revealed that all compounds are non-mutagenic except compound 12.” But in table 7 it says that compound 8 is also mutagenic.

-Regarding the experimental section

In the general procedure (line 443) says that “(…) few drops of glacial acetic acid”. The amount of acetic acid used should be specified (1 drop is considered to be 0.05 mL).

The mass spectrometry only has two decimals? Usually there are four, and the theoretical value obtained is also given.

Please note that in the 1H NMR of compound 8 there seem to be an extra hydrogen (24H instead of 23).

The 13C NMR of compound 12 is missing (in the article and in the supplementary data). Also, in the supplementary data, the 1H NMR spectra is very bad. If is not due to low amount of product (the yield was of 61%) is it problem of its solubility? And if it is the case, was the low solubility a problem for the activity assays?

The purity of synthesized compounds should be indicated.

-It would be better if some of the experimental information of the biological assays and of the in silico studies is in the article, instead of putting it all in the supplementary data. And the references that aren’t in the article should be placed in the supplementary data.

Kind regards

Author Response

Reviewer 3

In this article the authors discuss the design, synthesis and evaluation of five nicotinamide derivatives. Also, they made some in silico studies such as, docking, ADMET parameters and molecular dynamics.I have some comments about the paper:

Thank you for your efforts and your valuable comments. All comments were considered in high interest and all changes were highlighted in the revised manuscript.

  1. Why were the cellular lines HCT-116 and HepG-2 chosen for the assays? Have these cellular lines specially overexpressed VEGRF-2?

Response: VEGRF-2 is overexpressed in HCT-116 and HepG-2 cell lines. Accordingly, these two types of cells were utilized in our work. This fact was added into the revised manuscript.

  1. The schemes about the synthetic routes (schemes 1 and 2) aren’t detailed enough. Some conditions (temperature, time of reaction, …) and yields are missing. Also, ethanol is EtOH not EthOH.
  2. Response: Done
  3. If in the design proposed, there is a hydrophobic tail, why didn’t the authors use an aliphatic aldehyde? Also, it is also interesting to use and aldehyde with an CF3group (due to its pharmaceutical properties and to have some similarity with sorafenib).
  4. Response: thank you for this recommendation and this will be done in future work.
  5. In table 1, the SEM deviations of the IC50in the VEGFR-2 are missing. Also, it should state the number of times the experiments were performed (In the Supplementary data only says that the experiments were performed in triplicate for the PCR).
  6. Response: Done
  7. The explanation of figures captions and tables should be more elaborated.
  8. Response: Additional explanations were added in the captions of table and figures.
  9. Concerning the molecular docking: The versions of software used are from several years ago (MOE.14,…) why haven’t authors used more actual versions which have been improved?.
  10. Response: you are right regarding that is necessary to use more recent version of software. Unfortunately, MOE.2014 was the only version which is available in our institution. So that, we performed the docking studies using this version. We hope to use an updates version the future work.
  11. If they are using MOE for the docking, why have authors used other visualizers if it could be done with MOE? Also, some figures (2D image of figure 5) were obtained with MOE and others with other visualizers, is better if all the figures are obtained with the same software.
  12. Response: since Discovery studio visualizer give better figures, we used it as a visualizer for the results of MOE. In addition, Discovery studio visualizer can display the hydrophobic interaction in a good manner. This visualization process just displays the result in a good resolution. At the same time does not affect the results of docking in any way. If it is necessary to display these figures using MOE, we can d it.
  13. The molecules in figures 5, 6, 7 and 8 have different orientations. They should all have the same orientations in order to compare them and to easily locate the residues and interactions.

Response: in general, the features of all compounds at fig. 5, 6, 7, and 8 occupied the same pocket in the receptors as clarified in the discussion of the docking studies. You are right regarding the different orientation between the 3D and 2D figures. We cannot change these orientations as the software display the 2D in an orientation which is not necessarily similar to that of 3D. In addition, in each 3D figure, we tried to display all amino acids and essential interactions.  

  1. In the article the PDB used is 4ASD, but inthe Supplementary data there are two PDB listed. Why was the PDB:4ASD selected?
  2. Response: we used 4ASD protein because the co-crystallized ligand of this protein is Sorafenib and this more relevant to our study as sorafenib was used as a reference molecule in the biological testing. Accordingly, we used 4ASD to match with biological section regarding the reference molecule. The other protein (2OH4) involves a co-crystallized ligand may not related to the synthesized compound and we cannot use it a refence in the biological tests. Moreover, we deleted the other one from the supplementary data.

  1. In figure 5: Of the two molecules present in the figure (red and green) it should be address which is the re-docked conformer and which one is the co-crystalized conformer. Also, in the 2D image there is no alienation of the two molecules (as it is in the 3D image).

Response: each molecule was addressed in the caption of fig. 5. In addition, another pose which give a better alignment was added in the revised manuscript.

  1. When the protein was being prepared for the docking, authors removed all the water molecules. But, there is a water molecule (H2O 2090) that has interactions with Leu840 and with a carbonyl group from sorafenib.

Response: yes, there is a water-mediated hydrogen bond with Leu840, but this interaction is not essential for minimum affinity with the receptor. Accordingly, we deleted all water molecules from the protein. At the same time and if you persist, we can redock the synthesized compound against the protein structure conserving the water2090.

  1. The binding free energies (ΔG) are referred as docking score, but the “S score” should be also given.

Response: done

  1. Amino acids Cys919, Glu885 and Asp1046 are referred to as essential and key amino acids. The reason of this should be discussed.

Response: additional explanation was added in the docking section

  1. Some residues are different in the text than in the figures. For example, in line 306 it says “Cys1945” but in figure 6 it appears “Cys1045”. In line 321 says “Glu883” but in figure 7 it appears “Glu885”, in line 322 says “Asp1044” but in figure 7 it appears “Asp1046”.

 Response: corrected

  1. Also, the hydrophobic interactions in residues that don’t have aromatic rings and thus that don’t form p-p stacking aren’t represented.

Response: done 

  1. In figures 6, 7 and 8 there are three images in each (surface, 3D and 2D binding modes) but they are repetitive and show nothing different. With one image that show the interactions is enough.

Response: two images from each three were transferred into supplementary data.

  1. One figure that could be interesting is a superposition of sorafenib and compound 8.

Response: Done

  1. In line 298 it says “(…) the most cytotoxic derivatives 7 and 8 were selected for analyzing their binding 298 interactions and orientations.” But compound 7 is the third most active with respect to VEGFR-2 (with an IC50of 112.6 nM, compound 11 is the second with an IC50 of 83.41 nM) and the forth respect the two cellular lines (compounds 8, 6 and 12 are more active).

Response: Thank you for this notice. The sentence was corrected to be compatible with the results.  

  1. With respect to the in silicoADMET properties: Why are the parameters calculated with other software (Discovery Studio 4.0)? It could be done with MOE.

Response: ADMET studies were carried out using Discovery studio since it gives a chart for the results which are absent in MOE. In addition, we used to utilize Discovery studio to perform ADMET studies in an efficient manner. However, one of our team should be trained to perform ADMET studies using MOE.

  1. If the oral distribution is going to be discussed (table 7), the rule of five of Lipinsky could be addressed and if the compounds have any violations of these rules.

Response: Lipinski’s Rule of five and Veber's Rule were conducted in the revised work. There not violations of Lipinski’s Rule. Only compound 12 showed high total polar surface area violating Veber Rule.

  1. In line 352 says: “The results revealed that all compounds are non-mutagenic except compound 12.” But in table 7 it says that compound 8 is also mutagenic.

Response: thank you for this notice. This was corrected in the revised manuscript.

  1. Regarding the experimental section. In the general procedure (line 443) says that “(…) few drops of glacial acetic acid”. The amount of acetic acid used should be specified (1 drop is considered to be 05 mL).

Response: Done

  1. The mass spectrometry only has two decimals? Usually there are four, and the theoretical value obtained is also given.

Response: four decimals were added for mass spec. of theoretical values

  1. Please note that in the 1H NMR of compound 8 there seem to be an extra hydrogen (24H instead of 23).

Response: thank you for this notice. This protein was adjusted as appeared in supp data (23H)

  1. The 13C NMR of compound 12 is missing (in the article and in the supplementary data). Also, in the supplementary data, the 1H NMR spectra is very bad. If is not due to low amount of product (the yield was of 61%) is it problem of its solubility? And if it is the case, was the low solubility a problem for the activity assays?

Response: 13C NMR of compound 12 was added in the revised manuscript. 1H NMR of compound 12 was repeated and the new chart was added in the revised supp data.

  1. The purity of synthesized compounds should be indicated.

Response: As appeared in NMR, there is any noise in the base line indicating the high purity of the synthesized compounds. Carrying HPLC analyses will take a long time. If you persist, we can do it with given an extra time.

  1. It would be better if some of the experimental information of the biological assays and of the in silicostudies is in the article, instead of putting it all in the supplementary data. And the references that aren’t in the article should be placed in the supplementary data.

Response: We moved the experimental part to the supp. Data to keep the similarity (Plagiarism) level less than 20%.

It’s a serious consideration for our reputation as authors as well as for the journal.

Reviewer 4 Report

The reviewed manuscript describes designing, synthesis and biological study of a new VEGFR-2 inhibitors. Authors prepared five new analogs of Sorafenib. One of them (compound 8) exhibited very interesting activity against HCT-116 and HepG2 cancer cell lines. The same compound was the best VEGFR-2 inhibitor. Additionally, the effect of derivative 8 on the cell cycle,  level of the TNF-α and IL-6 proteins and on the expression levels of caspase-8 and BAX were examined. Reviewed manuscript present new and interesting biological results. Therefore, I would recommend the publication of reviewed manuscript in Molecules after minor revisions according to the following comments:

- at page 5 (figure 2) the target compounds are numbered 7a-c, while at page 7 (scheme 1) the same compounds are numbered 6-8. Please clarified it.

-page 23/line 451 (1H NMR of the compound 6): the signals at ca. 7.9-8.0 ppm  are doublet (at  8.04 ppm, 1H) together with two doublets (with total integration 4H) of AB spin system of the para substituted phenyl ring (see SI file, page 10), not doublet of triplets as it is describe in the manuscript (furthermore, the coupling constant J=25.5 Hz is “absurdly” high).

-page 23/line 460 (1H NMR of the compound 7): the signal at ca. 7.97 ppm is not doublet.  Similarly to the above comment, there are two doublets (with total integration 4H) of AB spin system of the para substituted phenyl ring (see SI file, page 14).

-Comments to the NMR spectra of the compound 11: first of all, this compound exists as a mixture of E/Z isomers, not as a pure E isomer as it state at page 24 (line 452). The signals of the second isomer are clearly indicated on both NMR spectra (pages 20-21of SI file). Thus, 1H NMR spectrum have to be reintegrated (please integrate all signals and calculate the molar ratio of both isomer).  Furthermore, there is lack of 13C NMR spectrum (spectrum presented on page 21 of SI file is an dept135 spectrum, not 13C NMR). Please add the copy of 13C NMR spectrum to the SI file. Finally, description of both spectra (page 24, line 484-488) should be complemented with the data of 1H and 13C NMR of the second, minor isomer.

-Comments to the NMR spectra of the compound 12: description of the 1H NMR spectrum (page 24/line 491-493) does not correspond to the copy of 1H NMR spectrum presented in the SI file (page 23). Moreover, the quality of the mentioned copy of the 1H NMR spectrum is unacceptable – all signals are very broad (it could be caused by paramagnetic impurities or wrong shimming). Thus, 1H NMR spectrum of compound 12 have to be measured once more. Furthermore, there is lack of 13C NMR spectrum – this spectrum have to be also measured and appropriate data have to be added to the main text of manuscript and to the SI file.

Author Response

Reviewer 4

The reviewed manuscript describes designing, synthesis and biological study of a new VEGFR-2 inhibitors. Authors prepared five new analogs of Sorafenib. One of them (compound 8) exhibited very interesting activity against HCT-116 and HepG2 cancer cell lines. The same compound was the best VEGFR-2 inhibitor. Additionally, the effect of derivative 8 on the cell cycle,  level of the TNF-α and IL-6 proteins and on the expression levels of caspase-8 and BAX were examined. Reviewed manuscript present new and interesting biological results. Therefore, I would recommend the publication of reviewed manuscript in Molecules after minor revisions according to the following comments:

Thank you for your efforts and your valuable comments. All comments were considered in high interest and all changes were highlighted in the revised manuscript.

  1. At page 5 (figure 2) the target compounds are numbered 7a-c, while at page 7 (scheme 1) the same compounds are numbered 6-8. Please clarified it.

Response: corrected

  1. page 23/line 451 (1H NMR of the compound 6): the signals at ca. 7.9-8.0 ppm  are doublet (at  8.04 ppm, 1H) together with two doublets (with total integration 4H) of AB spin system of the para substituted phenyl ring (see SI file, page 10), not doublet of triplets as it is describe in the manuscript (furthermore, the coupling constant J=25.5 Hz is “absurdly” high).
  2. Response: Thank you for this notice. The data were adjusted as appeared in the supplementary materials.
  3. page 23/line 460 (1H NMR of the compound 7): the signal at ca. 7.97 ppm is not doublet.  Similarly to the above comment, there are two doublets (with total integration 4H) of AB spin system of the para substituted phenyl ring (see SI file, page 14).
  4. Response: Thank you for this notice. The data were adjusted as appeared in the supplementary materials.
  5. Comments to the NMR spectra of the compound 11: first of all, this compound exists as a mixture of E/Z isomers, not as a pure E isomer as it state at page 24 (line 452). The signals of the second isomer are clearly indicated on both NMR spectra (pages 20-21of SI file). Thus, 1H NMR spectrum have to be reintegrated (please integrate all signals and calculate the molar ratio of both isomer).  Furthermore, there is lack of 13C NMR spectrum (spectrum presented on page 21 of SI file is an dept135 spectrum, not 13C NMR). Please add the copy of 13C NMR spectrum to the SI file. Finally, description of both spectra (page 24, line 484-488) should be complemented with the data of 1H and 13C NMR of the second, minor isomer.

Response: 13C NMR of compound 11 was added in the revised supp data.

  1. Comments to the NMR spectra of the compound 12: description of the 1H NMR spectrum (page 24/line 491-493) does not correspond to the copy of 1H NMR spectrum presented in the SI file (page 23). Moreover, the quality of the mentioned copy of the 1H NMR spectrum is unacceptable – all signals are very broad (it could be caused by paramagnetic impurities or wrong shimming). Thus, 1H NMR spectrum of compound 12 have to be measured once more. Furthermore, there is lack of 13C NMR spectrum – this spectrum have to be also measured and appropriate data have to be added to the main text of manuscript and to the SI file.

Response: new 1H NMR and 13C NMR analyses of compound 12 were added in the revised manuscript and revised supp data.

Round 2

Reviewer 1 Report

Deras,

the manuscript is well improved.

Best wishes 

Author Response

Thank you for your valuable efforts

Reviewer 2 Report

After the revision, I recommend the manuscript for publication. Only minor spell check and English editing should be done.

Author Response

Thank you for your valuable efforts. All comments were considered in high interest and all changes were highlighted in the revised manuscript.

Reviewer 3 Report

I was again invited to review the manuscript “Design, synthesis, in silico and in vitro studies of new immunomodulatory anticancer nicotinamide derivatives targeting VEGFR-2” based on a previous submission. Some aspects of my previous review were followed and addressed. However, there are still some issues:

-If authors prefer to use the Discovery visualizer, all figures should be done with the same program. For example, figures 5 and 6 are different/done with different programs.

-Regarding the mass spectrometry: In the article is given like (for example): “C20H14Cl2N4O2 (413.2580g/mol)” but the value given is the experimental or the theoretical one? (when reporting the mass data, authors may give both I order to compare the deviation).

-In the 1H NMR of compound 11 in the supplementary data, some “extra signs” can be seen. Thus, the purity is not very high. This is one of the reasons why the purity of synthesized compounds should be indicated.

-The 13C NMR of compound 12 is not acquired enough (needs more time), the signals of the quaternary carbons couldn’t be seen (there is too much noise). Also, in the 1H NMR all the signals seem to be broad singlets, I don’t get how could it be possible to determine the multiplicity and obtain the constants.

-If this issues with NMR is not due to low amount of product (the yields are good) is it problem of the solubility? And if it is the case, was the low solubility a problem for the activity assays?

Author Response

I was again invited to review the manuscript “Design, synthesis, in silico and in vitro studies of new immunomodulatory anticancer nicotinamide derivatives targeting VEGFR-2” based on a previous submission. Some aspects of my previous review were followed and addressed. However, there are still some issues:

Thank you for your valuable efforts. All comments were considered in high interest and all changes were highlighted in the revised manuscript.

  • If authors prefer to use the Discovery visualizer, all figures should be done with the same program. For example, figures 5 and 6 are different/done with different programs.

Response: All figures are visualized by Discovery studio.

  • Regarding the mass spectrometry: In the article is given like (for example): “C20H14Cl2N4O2 (413.2580g/mol)” but the value given is the experimental or the theoretical one? (when reporting the mass data, authors may give both I order to compare the deviation).

Response: the given mass are the theoretical values. We did not carry out experimental HRSM.

  • In the 1H NMR of compound 11 in the supplementary data, some “extra signs” can be seen. Thus, the purity is not very high. This is one of the reasons why the purity of synthesized compounds should be indicated.

Response:   As commended before from reviewer-4, compound 11 is a mixture of E and Z forms. Accordingly, there is a duplication for some signal. In addition, the base line of 1H NMR charts of the synthesized are clear indicating the high purity of these compounds. To carry out purity test as elemental analysis, we need long time. If you persist, we will do it.

  • The 13C NMR of compound 12 is not acquired enough (needs more time), the signals of the quaternary carbons couldn’t be seen (there is too much noise). Also, in the 1H NMR all the signals seem to be broad singlets, I don’t get how could it be possible to determine the multiplicity and obtain the constants.

Response:  The 13C NMR of compound 12 was carried twice. You are right that this analysis needs more extra time to give clearer chart. However, we can detect the peak of carbons from the current chart. We can not repeat this analysis for the third time. For the 1H NMR of compound 12, I see that is very acceptable since we can deduce the splitting easily. There is no need to repeat it since it will give the same results. Hope you understand our situation that repeating these analyses for three time is very difficult.

  • If this issues with NMR is not due to low amount of product (the yields are good) is it problem of the solubility? And if it is the case, was the low solubility a problem for the activity assays?

Response: We did not have any problem related to yield or solubility.